# Transcriptomic Analysis Provides Insights into Candidate Genes and Molecular Pathways Involved in Growth of *Mytilus coruscus* Larvae

**DOI:** 10.3390/ijms25031898

**Published:** 2024-02-05

**Authors:** Minhui Xu, Zhong Li, Xinjie Liang, Jiji Li, Yingying Ye, Pengzhi Qi, Xiaojun Yan

**Affiliations:** National Engineering Research Center for Marine Aquaculture, Zhejiang Ocean University, Zhoushan 316022, China; xuminhui@zjou.edu.cn (M.X.); 18103718720@163.com (Z.L.); liangxinjie@zjou.edu.cn (X.L.); lijiji@zjou.edu.cn (J.L.); yanxj@zjou.edu.cn (X.Y.)

**Keywords:** *Mytilus coruscus*, larvae, transcriptomic, growth genes, molecular pathways

## Abstract

Growth is a fundamental aspect of aquaculture breeding programs, pivotal for successful cultivation. Understanding the mechanisms that govern growth and development differences across various stages can significantly boost seedling production of economically valuable species, thereby enhancing aquaculture efficiency and advancing the aquaculture industry. *Mytilus coruscus*, a commercially vital marine bivalve, underscores this importance. To decipher the intricate molecular mechanisms dictating growth and developmental disparities in marine shellfish, we conducted transcriptome sequencing and meticulously analyzed gene expression variations and molecular pathways linked to growth traits in *M. coruscus*. This study delved into the molecular and gene expression variations across five larval development stages, with a specific focus on scrutinizing the differential expression patterns of growth-associated genes using RNA sequencing and quantitative real-time PCR analysis. A substantial number of genes—36,044 differentially expressed genes (DEGs)—exhibited significant differential expression between consecutive developmental stages. These DEGs were then categorized into multiple pathways (Q value < 0.05), including crucial pathways such as the spliceosome, vascular smooth muscle contraction, DNA replication, and apoptosis, among others. In addition, we identified two pivotal signaling pathways—the Hedgehog (Hh) signaling pathway and the TGF-beta (TGF-β) signaling pathway—associated with the growth and development of *M. coruscus* larvae. Ten key growth-related genes were pinpointed, each playing crucial roles in molecular function and the regulation of growth traits in *M. coruscus*. These genes and pathways associated with growth provide deep insights into the molecular basis of physiological adaptation, metabolic processes, and growth variability in marine bivalves.

## 1. Introduction

Over the last few decades, aquaculture has surged ahead as the fastest-expanding sector in the global food industry, boasting an impressive average annual growth rate of 8% since the 1970s [1]. Mussels, cultivated extensively worldwide, contribute significantly to this growth, yielding a total production of 1.205 million tons [2]. In China, the fourth largest farmed shellfish in the aquaculture industry, mussels account for 886,875 tons of annual production [1]. The thick-shelled mussels (*Mytilus coruscus*) (Mollusca, Bivalvia, Mytiloida, Mytilidae) are important species distributed in the coastal waters of China (i.e., Bohai Sea, Yellow Sea, and Taiwan’s Penghu Islands) and resides in subtidal zones, attaching to hard substrates and providing food and habitat for numerous other species. They are commonly referred to as “dan cai”, and as significant marine bivalve species with important economic and ecological value and a variety of nutrients that are vital to humans [2,3]. Moreover, the genus *Mytilus* exhibits powerful tolerance to varying environmental conditions, making them valuable indicators of environmental quality in coastal waters [4]. *M. coruscus*, with its rapid growth cycle and straightforward cultivation, boasts unique attributes that render it a significant mollusca species in our country’s coastal aquaculture sector. The proportion of *M. coruscus* aquaculture has been progressively increasing, transforming it into one of the key pillar industries in coastal regions of our nation.

Embryonic and larval development in bilaterals is an important phylogenetic process. Similar to most marine invertebrates, the life history of *M. coruscus* comprises two distinct stages: a planktonic phase consisting of trochophore larvae, D-veliger larvae, veliconcha larvae, and pediveliger larvae, followed by a semi-attached benthic phase involving juveniles [5]. The growth and development quality during the larval stage directly influences the subsequent reproductive capacity and survival success rate. Additionally, the adaptive capabilities during the larval stage are of great significance for population maintenance and reproduction. Thick-shelled mussel larvae encounter a medley of environmental pressures and challenges during this phase, including changes in temperature, salinity, nutrients, and light conditions. Only with sufficient adaptability can they survive and reproduce under different environmental conditions. Nonetheless, aquaculture of *M. coruscus* confronts an array of risks and constraining factors, akin to other animal production endeavors, such as environmental shifts, pollution, and the threat of infectious diseases [4,6].

In recent years, transcriptomics has rapidly advanced in aquatic biology research. RNA-seq, an influential method for unraveling molluscan transcriptomes, has gained considerable attention. This approach involves sequencing the entire repertoire of RNA molecules within a sample, enabling researchers to measure gene expression levels, unveil novel genes, and detect alternative splicing events. Moreover, RNA-seq allows for the exploration of non-coding RNAs, including microRNAs and long non-coding RNAs, which hold pivotal roles in gene regulation and developmental processes [7,8].

Previous studies on Mollusca transcriptomics have yielded significant achievements and discoveries. They shed light on the molecular mechanisms underpinning vital biological processes like shell formation, growth, immune responses, and environmental adaptation. Key genes and pathways related to shell matrix proteins, enzymes, and transporters crucial in biomineralization processes have been identified through transcriptomic studies [9,10]. Moreover, transcriptomic analyses have identified immune-related genes, signaling pathways, and defense mechanisms against pathogens. Moreover, studies have proposed that mTOR plays a conserved role in controlling protein synthesis, glucose metabolism, lipid synthesis, and autophagy in razor clam (*Sinonovacula constricta*) [11] and is related to the linear relationships with growth rates in bivalve larvae (*Crassostrea gigas*) [12]. Thus, mTOR may be an attractive target for promoting bivalve growth. Glebov et al. found that 5-HT is involved in the regulation stage transition in *Helisoma trivolvis* including larval metamorphosis and settlement [13]. Nie et al. identified the TGF-beta signaling pathway, MAPK signaling pathway, and notch signaling pathway, which influenced the growth rate of Manila clam [14]. Overall, these theoretical bases have significantly enriched our comprehension of the molecular foundations governing critical biological processes and the genetic mechanisms underlying the distinctive attributes of Mollusca species.

Presently, extensive studies exist concerning genes pertinent to growth and development primarily within vertebrates and insects [15,16,17,18]. However, there is a notable scarcity of studies elucidating embryogenesis and larval development in this species and related ones. Understanding these processes, especially the timing of developmental stages, is vital for advancing hatchery techniques for this economically significant species. In our study, we have previously sequenced and characterized the *M. coruscus* transcriptome across five distinct developmental stages. Leveraging RNA sequencing and quantitative real-time PCR methods, we examined the expression patterns of two signaling pathways and key genes involved in growth and development. Our primary objective was to identify differentially expressed genes and understand their expression patterns, enriching our understanding of the genetic foundations and transcriptomic regulations guiding growth differentiation in thick-shelled mussel larvae. Furthermore, we aimed to probe growth-related genes and molecular pathways associated with growth and development regulation in *M. coruscus* larvae. This knowledge not only enhances survival rates in economically vital *M. coruscus* but also bolsters aquacultural efficiency, propelling progress within the aquaculture sector.

## 2. Results

### 2.1. Data Quality Control and Global Gene Expression Analysis

Statistical methods were applied to assess and manage the measured sequences, providing a comprehensive overview of library construction and sequencing quality. As presented in Table 1, following quality control of the sequencing results, each sample exhibited a Q30 percentage exceeding 90%, indicating highly accurate sequencing base quality. When comparing the clean reads with the mussel’s reference genome, clean reads accounted for more than 95% of the raw reads.

Performing principal component analysis (PCA) conducted on the five larval developmental stages effectively segregated all 15 sample pools into five distinct groups (Figure 1). The first and second principal components collectively explained 50.4% of the variation in the entire dataset, suggesting that veliconcha, pediveliger, and juvenile samples exhibited greater similarities compared to the preceding two stages. It was found that the larvae of the same developmental stages were clustered into one group each, enhancing the reliability and repeatability of this study. To cluster protein expression profiles across all developmental stages, we employed the fuzzy c-means algorithm [19]. This analysis identified 12 distinct clusters of temporal patterns, representing proteins with differing regulation patterns (Figure 2; Appendix A). Notably, clusters 1 and 4 exhibited significant upregulation during the trochophore stages, while clusters 7, 11, and 12 showed significant upregulation during the juvenile stages. Clusters 2, 3, 8, and 9 displayed upregulation with stage-specific patterns across three larval stages, while clusters 5, 6, and 10 exhibited a bi-modal expression pattern. These time course clusters provide insights into the sequence of action for differentially regulated yet co-expressed genes. A total of 21,521 differentially expressed genes (DEGs) were shared in five different stages (Figure 3a) and significantly enriched in five KEGG pathways including apoptosis, thermogenesis, spliceosome, the insulin signaling pathway, and lysosome (Figure 3c). GO analysis, on the other hand, showed that the DEGs enriched cellular anatomical entity, binding, and catalytic activity (Figure 3c).

### 2.2. Transcriptional Changes across Larval Stage Transitions

In this study, we used DESeq2 (repeated samples) to analyze the significant gene differences in *M. coruscus* across consecutive developmental stages. Among the extensive pool of 36,044 unigenes, the most pronounced variation in gene expression was observed between trochophore and D-veliger stage larvae, yielding 11,012 significant genes. Conversely, the fewest genes exhibited alterations between the veliconcha and pediveliger stages, amounting to 5009 significant genes. The majority of genes displayed significant alterations in expression profiles during larval development (Figure 4).

### 2.3. KEGG and GO Enrichment Analyses of Differential Gene Expression Profiles under Comparison of Two Consecutive Stages

In our analysis, significant differences in GO terms (Appendix A) and KEGG pathways (Appendix A) were evident between consecutive developmental stages (Q value < 0.05). Between trochophore and D-veliger stage larvae, we identified 65 significantly enriched KEGG pathways. The majority were associated with genetic information processing, organismal systems, and cellular processes (e.g., ko03040://Spliceosome, ko04270://Vascular smooth muscle contraction, ko04210//Apoptosis, ko04142//Lysosome). For the D-veliger-to-veliconcha stage transition, 58 KEGG pathways were significantly enriched, mainly linked to organismal systems (e.g., ko04714//Thermogenesis, ko04270://Vascular smooth muscle contraction). Comparatively, only 15 KEGG pathways showed significant enrichment between veliconcha and pediveliger stage larvae, primarily associated with genetic information processing (e.g., ko03010://Ribosome, ko03040://Spliceosome).

In the GO term analysis, GO:0003824//catalytic activity emerged as the predominant term in molecular function. Moreover, when comparing pediveliger stage larvae to juveniles, 65 KEGG pathways exhibited significant enrichment. Across these comparisons, genes that underwent significant changes were chiefly associated with GO:0003824//catalytic activity in the molecular function GO term analysis.

### 2.4. Growth-Related Genes and Molecular Pathways in M. coruscus Larvae and qPCR Validation

According to this research, numerous growth-related pathways were identified, KEGG analysis unveiled significant molecular pathways such as the Hedgehog signaling pathway and TGF-beta signaling pathway (Figure 5), both known to promote cell proliferation, metabolism, and protein synthesis. These findings strongly suggested that these pathways potentially play vital roles in growth regulation, cell growth, differentiation and development, organizational homeostasis, and embryonic development of *M. coruscus*.

The results exhibited a consistent expression trend with those obtained from RNA-seq analyses, as depicted in Figure 6 and Figure 7. However, variations in expression levels were noted, likely attributable to differing detection sensitivities between transcriptome sequencing and qRT-PCR experiments, as well as disparities in analysis methods.

## 3. Discussion

The development of and growth in larvae processes were closely related to the regulation of gene expression and the transcriptome. Understanding the growth and development of larvae can help optimize *M. coruscus* culture techniques and help improve their growth rate and survival rate. Studies have demonstrated the fundamental roles of the Hedgehog signaling pathway and transforming growth factor-β in normal embryonic development, emphasizing their pivotal involvement in intercellular signaling, as well as the maintenance, renewal, and regeneration of adult tissues [20,21]. *Shh* (sonic hedgehog homology) binding to *Ptc* (patched) activates *Smo* (smoothened), causing dissociation of *SuFu* from *Gli* and forming *GliA*, promoting the expression of target genes. Evidence from several studies suggested that *Shh* controlled early vertebrate midline organization, control of embryonic left and right, and the establishment of the dorsoventral axis [22,23]. The results of this study showed that the expression of *Shh* was basically absent in the period of trochophore and veliconcha stages, and less in the period of D-veliger and pediveliger stages. The embryonic shell begins to form in the D-veliger stage and the shell morphology and structure of the larva change during the transition from the D-veliger to veliconcha stages [24]. The study reported that the expression of the *Smo* gene promoted the proliferation of mouse chondrocytes [25], whereas mutations in *Smo* caused phenotypic defects in zebrafish body size and cartilage [26]. Cynthia proposed that mice without specific gene *Ptc* influenced the *Shh*/*Ptc* pathway on both normal and abnormal cell proliferation [27], indicating the functional consequences of its inactivation on the control of proliferation. Based on this study, *Smo* was expressed at a lower level in the first four periods, while Ptch1 and *Gli2/3* showed a gradual increase in expression as downstream transcription factors of the pathway. We suggested that these might be involved in changes in the morphological structure of shells. All genes involved in the Hedgehog signaling pathway exhibited significant upregulation during the juvenile stage. In Mollusca, studies have explored the involvement of the Hedgehog signaling pathway in myogenesis, as observed in *Sepia officinalis* [28] and *Crassostea gigas* [29]. Specifically, Hedgehog signaling played a critical role in the precise patterning of obliquely striated muscle fibers in the cuttlefish mantle. In the juvenile stage, secondary shell formation occurs in *M. coruscus* larvae. The juveniles secrete byssal threads for attachment and sustenance, with well-developed gills enabling respiration and filter feeding. It was speculated that the genes in the Hedgehog signaling pathway may be involved in the attachment metamorphosis development of larvae, which promotes the formation of secondary shells in larvae through cell proliferation and may be involved in the physiology of larvae and the development of some organs.

The transforming growth factor-β (TGF-β) superfamily represents a fundamental and ancient mode of intercellular signaling in eukaryotic animals [30]. TGF-β signaling is pivotal in various developmental and homeostatic processes, including the establishment of the embryonic body plan and subsequent determination and maintenance of cell identities [31]. Within this superfamily, members like TGF-betas, activins, and BMPs (bone morphogenetic proteins) are structurally related secreted cytokines, found across a wide range of species from worms and insects to mammals [32,33]. The biological effects of BMPs are conveyed through intracellular pathways such as BMP–Smad and BMP–MAPK. Loss-of-function studies on TGF-β family ligands in mice emphasize their crucial roles in embryonic development and maintaining tissue balance into adulthood [34]. Notably, in pearl oysters and other bivalves, while the typical TGF-beta molecule is absent, essential components of this signaling pathway, including the primary growth factor BMPs, are still present [35]. *Smads*, acting as intracellular nuclear effectors of the TGF-β family, play a crucial role. The intracellular signaling triggered by *Bmps* initiates as the ligands bind to their transmembrane serine/threonine kinase receptors, leading to the phosphorylation of members of the *Smad* family. The *Smad2* gene is a downstream signaling molecule of the TGF-β superfamily; we analyzed the expression profiles and found that *Smad2/3* was upregulated continuously, which played the role of propagating signals in the whole larval developmental period. *Smd4* is involved in regulating various physiological processes such as tumor metastasis, chondrocyte and bone development, and oocyte growth through the TGF-β pathway [36,37,38]. Its high expression in the five stages of our study implied potential roles in *M. coruscus* development. The BMP protein family, which includes BMP2, BMP4, and BMP7, among others, constitutes bone morphogenetic proteins known as osteoinductive factors that induce osteoblast differentiation. *Bmp7*, *Bmpr1b,* and *Bmp2/4* upregulated from the trochophore to the D-veliger stage. At this stage, the primary shell of mussel larvae is starting to develop, and some organ primordia, like the primordia for the closing shell muscles, have begun to take shape [24]. In addition, *Bmp7* expression peaked during the veliconcha stage, while *Bmp2/4* exhibited the highest expression levels during the juvenile stage. The *M. coruscus* undergoes a morphological transformation from the D-veliger stage to the veliconcha stage. A notable aspect of this metamorphic process is the detachment of the larval velum, accompanied by the development of gills and the formation of the secondary shell [39].

The construction of shells in bivalves is a crucial aspect during the stages of settlement and metamorphosis. The formation of primary and secondary shells can provide protection for the larvae. Additionally, the transition in lifestyle from being planktonic to settling at the bottom is important to larvae. Currently, the seedlings of *M. coruscus* abalone are primarily sourced from artificial breeding. However, artificial breeding technology is not yet mature, with less than 10% of *M. coruscus* larvae successfully undergoing metamorphosis each year. There is an urgent need to improve the metamorphosis rate in artificial breeding. These findings confirmed that genes contribute to the development and growth of bone and muscle, associated with the development of organs like the primary shell and gills, suggesting that these genes were an attractive candidate gene for the selection of development and growth related to *M. coruscus*. In response to this, we can utilize technologies such as gene editing to enhance the survival rates of *M. coruscus* larvae at different developmental stages, thereby fostering the cultivation of high-quality seedlings. Due to the small size of *M. coruscus* larvae, technical challenges arise in the implementation of RNA interference and similar techniques. Furthermore, the specific mechanism remained to be further explored.

## 4. Materials and Methods

### 4.1. Larvae Collection

The *M. coruscus* individuals used during this experiment were provided by the Shengsi Marine Science and Technology Institute, Zhejiang province (30°44′40.91″ N, 122°27′57.45″ E). The larvae samples of five periods of *M. coruscus* were collected, including 1 dpf (days post fertilization), 3 dpf, 15 dpf, 35 dpf, and 60 dpf (trochophore, D-veliger, veliconcha, pediveliger, and juvenile stages). About the collection of samples of mussels at each developmental stage, 500 μL larvae samples stored in RNAlater solution (Solarbio, Beijing, China) were absorbed and about 30 larvae were placed under a 10× microscope (different sizes of larvae at each stage were collected after density was adjusted in the breeding farm). Adequate larval samples were collected at each stage for total RNA extraction and stored at −80 °C for subsequent analysis.

### 4.2. RNA Isolation, cDNA Library Preparation, and Sequencing

The extraction of total RNA from larval samples was performed following the manufacturer’s guidelines using TRIzol reagent (Invitrogen, Carlsbad, CA, USA). The integrity of the extracted RNA was assessed using NanoDrop 2000 (Thermo Fisher Scientific, Waltham, MA, USA) and 1% agarose gel electrophoresis [40]. The RNA was then sent to HuaDa BGI (Shanghai, China) to establish Illumina sequencing libraries, with three biological replicates sequenced for each tissue sample from every stage. To enrich mRNAs with polyA tails, magnetic beads with OligodT were employed. The obtained RNA was subsequently fragmented using an interrupting buffer, with 6-base random primers used to synthesize the first-strand DNA utilizing RNA as the template. The second-strand cDNA was synthesized using the initial first-strand cDNA as a template. The DNA ends were modified, flattened, and phosphorylated at the 5’ end, creating a sticky end with a protruding “A” at the 3′ end. A bulbous joint with a protruding “T” at the 3′ end was then ligated to this structure. Subsequently, specific primers were employed to amplify the ligation products through PCR. Following PCR, the resulting product was thermally denatured into a single strand. A circular DNA library was obtained by cyclizing the single-strand DNA using a bridge primer.

### 4.3. Data Analysis

The raw sequencing read quality before and after trimming was processed using SOAPnuke software (version v1.5.2) to remove reads containing adaptors, unknown nucleotides (base N) above 5%, and low-quality base reads with Q < 20. The filtered reads were saved in FASTQ format.

### 4.4. Differential Expression and Enrichment Analysis

We aligned the clean reads to the reference genome using HISAT (Hierarchical Indexing for Spliced Alignment of Transcripts) [41]. Subsequently, we utilized Ericscript (v0.5.5) [42] to pinpoint fusion genes, while identifying differential splicing genes (DSGs) was achieved through rMATS (V3.2.5) [43]. Briefly, clean reads were mapped to single genes using Bowtie 2 (version v2.3.0) [44] and read counts for each gene were calculated using the default settings of RSEM (http://deweylab.github.io/RSEM/ (accessed 6 November 2023)) (version v1.2.12) [45]. PCA was visualized using origin software (version 2022) [46]. The genes from the five stages were clustered using the Mfuzz package in R with the fuzzy c-means algorithm [47]. Subsequent analysis involved scrutinizing the variance in gene expression among samples using DEGseq, from which DEGs were identified. Moreover, we assessed the variance in gene expression among samples using DEGseq, and DEGs were singled out employing a threshold for adjusted Q value (Q_adj_ < 0.05) and a fold change > 2. To comprehend the phenotypic alterations, we conducted Gene Ontology (GO) and Kyoto Encyclopedia of Genes and Genomes (KEGG) enrichment analyses for annotated DEGs, employing TBtools (version v1.1.3.2) [48].

### 4.5. qRT-PCR Validation of the RNA-Seq Profiles

We selected 10 growth-related genes identified in the RNA-seq analysis for qRT-PCR validation based on the functional relevance of DEGs. The qPCR primers were designed according to the CDS sequence of the gene using Primer 5.0 software, and EF1A was the internal reference gene (Table 2). The qPCR was performed using the SYBR qPCR Mix kit (EnzyArtisan, Shanghai, China). The cDNA utilized as the qRT-PCR template was synthesized using the PrimeScript™ reverse transcription enzyme kit (Takara, Biotechnology, Kukatsu, Japan) and RNase inhibitor, adhering to the provided instructions. qRT-PCR was performed on an ABI-7500 rapid real-time system (Applied Biosystems, Foster City, CA, USA), and three biological replicates were set for each gene. The relative expression levels of the target gene were calculated using the optimized comparative Ct (2^−ΔΔCT^) method [49].

## 5. Conclusions

While genomics tools have gained traction in recent years for studying marine bivalves, there remains a notable dearth of research on the genetic data of *M. coruscus*. Previous studies have predominantly revolved around examining tissue, organ, and shell development, offering limited insights into variations in gene expression levels during larval development. In this study, we meticulously analyzed the transcriptome related to larval growth, utilizing an extensive transcript dataset. The resulting annotated transcripts represent invaluable genomic resources, providing a glimpse into the molecular underpinnings of growth characteristics. Through a comparative analysis of transcriptomes at different stages, we successfully pinpointed numerous differentially expressed genes likely associated with growth and development in this species. Additionally, our findings suggest the potential involvement of the Hedgehog signaling pathway and TGF-beta signaling pathway in crucial processes like cell differentiation, embryonic development, tissue homeostasis, and organ formation in *M. coruscus*, the related genes played different roles in different stages. This research significantly advances our understanding of pivotal genes and pathways governing the growth and development of *M. coruscus*, offering fresh perspectives on the molecular mechanisms underpinning growth differentiation in marine bivalves. However, further experiments are needed to verify the functions and regulatory mechanisms of the signaling pathway and related genes.

## Figures and Tables

**Figure 1 ijms-25-01898-f001:**
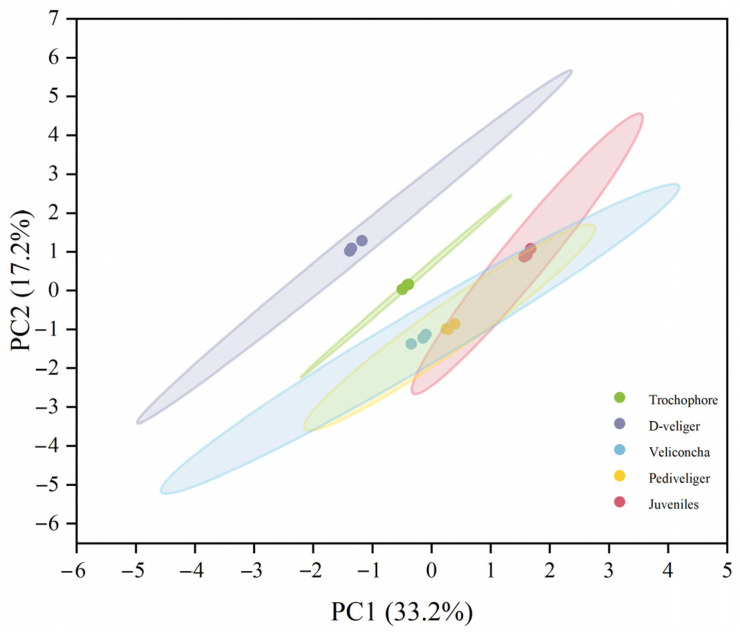
Two-dimensional score plot using the first 2PCs identified by principal component analysis on the entire larval gene set. Fifteen pools of samples were separated into five groups, relevant to the five larval developmental stages. Each stage point included a triplicate.

**Figure 2 ijms-25-01898-f002:**
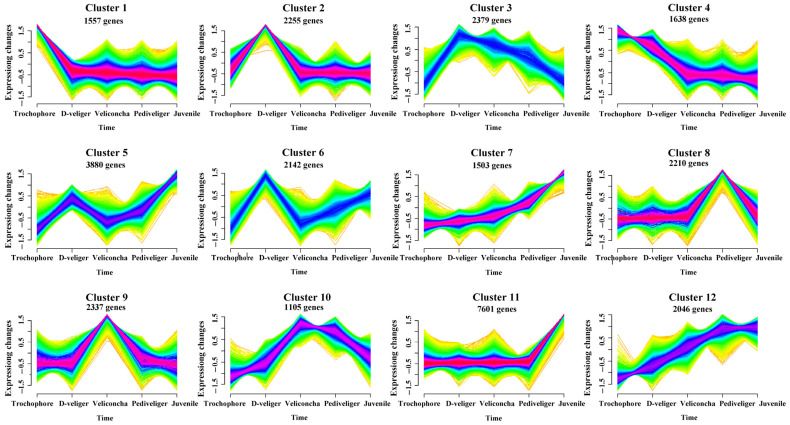
Fuzzy c-means clustering identified twelve distinct temporal patterns of gene expression. The *x* axis represents six developmental stages, while the *y* axis represents log2-transformed, normalized intensity ratios in each stage.

**Figure 3 ijms-25-01898-f003:**
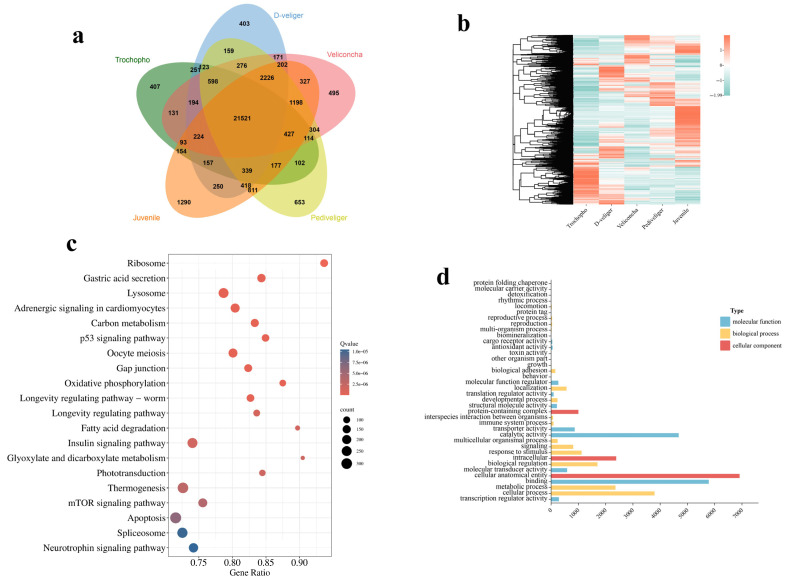
(**a**) Venn diagram of DEGs in five different stages. (**b**) Top 20 of KEGG enrichment of 21521 DEGs commonly regulated by five different stages. (**c**) GO categorization (biological process, cellular component, and molecular function) of 21521 DEGs in the transcriptome of *M. coruscus*.

**Figure 4 ijms-25-01898-f004:**
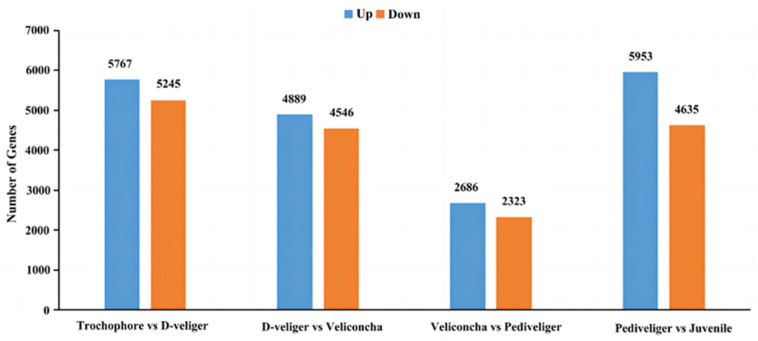
Transcriptional changes across larval stage transitions. The upregulated genes refer to those with FC > 2 while the downregulated genes refer to those with FC ≤ 0.5.

**Figure 5 ijms-25-01898-f005:**
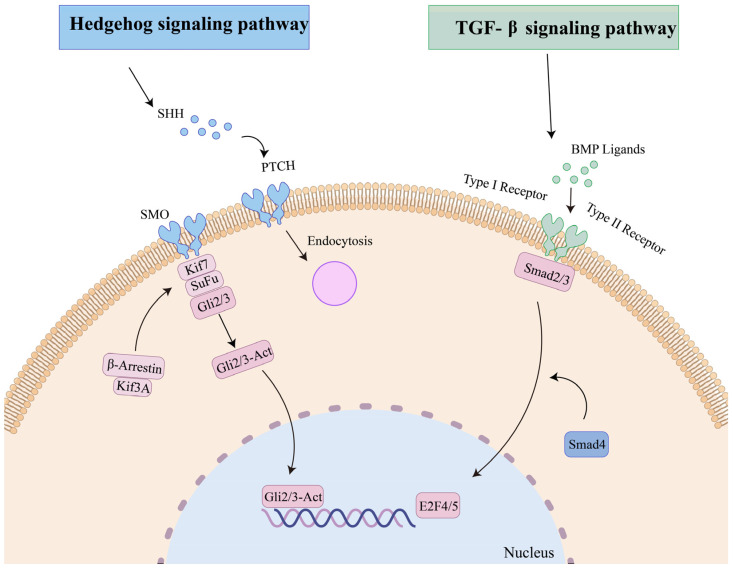
Molecular pathways identified in the KEGG analysis, including TGF-β signaling pathway, and Hedgehog signaling pathway.

**Figure 6 ijms-25-01898-f006:**
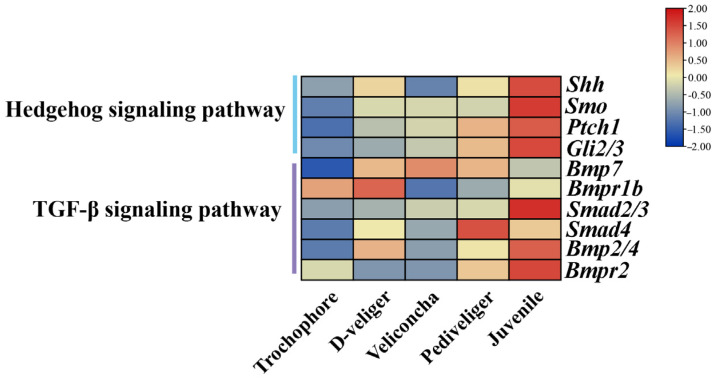
Heatmap showing changes in expression levels of genes in several growth-related pathways among five different stages mussel. The color scale represents gene expression (FPKM) normalized as *Z*-score, from blue with the lowest to red with the highest values. FPKM is a normalized estimation of gene expression based on RNA-seq data. The greater the FPKM value, the higher the gene expression level.

**Figure 7 ijms-25-01898-f007:**
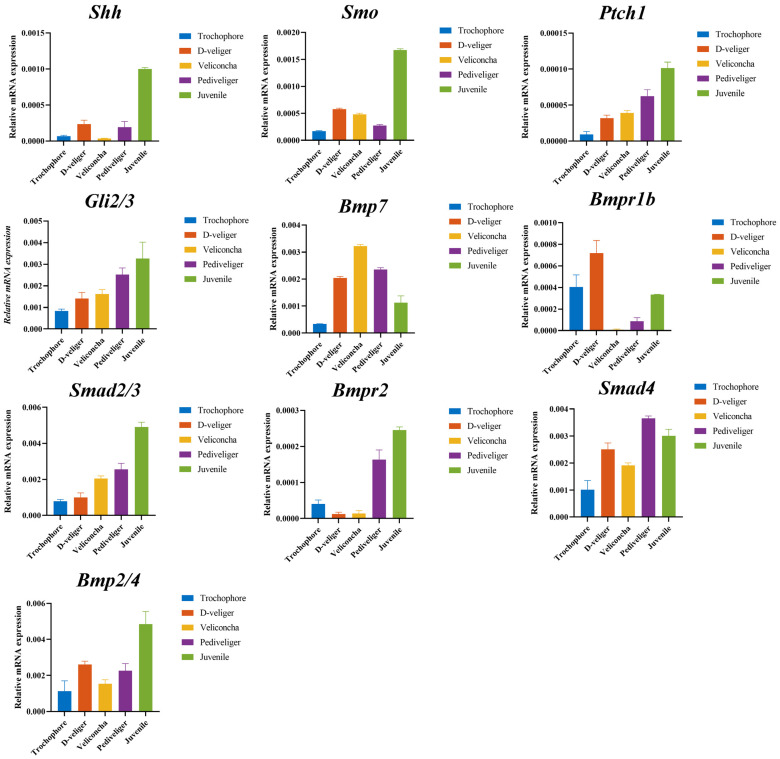
Expression patterns of 10 genes by real-time quantitative PCR analysis.

**Table 1 ijms-25-01898-t001:** The overview of transcriptome data of *M. coruscus*.

Sample	Total Raw Reads (M)	Total Clean Reads (M)	Total Clean Bases (Gb)	Clean Reads Q20 (%)	Clean Reads Q30 (%)	Clean Reads Ratio (%)
Trochophore1	43.82	42.16	6.32	96.53	91.37	96.21
Trochophore2	43.82	42.28	6.34	96.73	91.85	96.49
Trochophore3	45.57	42.42	6.36	97.01	92.51	93.08
D-veliger1	45.57	43.37	6.51	96.74	91.90	95.16
D-veliger2	43.82	41.99	6.30	96.78	91.91	95.82
D-veliger3	43.82	42.16	6.32	96.84	92.09	96.22
Veliconcha1	43.82	42.10	6.32	96.72	91.82	96.08
Veliconcha2	43.82	42.02	6.30	96.83	92.09	95.88
Veliconcha3	43.82	42.23	6.33	96.69	91.73	96.37
Pediveliger1	43.82	42.29	6.34	96.54	91.35	96.50
Pediveliger2	43.82	42.41	6.36	97.48	93.32	96.78
Pediveliger3	43.82	42.30	6.35	97.54	93.44	96.53
Juvenile1	43.82	42.19	6.33	97.24	92.81	96.28
Juvenile2	43.82	42.10	6.31	97.21	92.71	96.07
Juvenile3	43.82	42.03	6.30	97.19	92.70	95.91
Summary	660.80	634.05	95.09	Mean = 96.94	Mean = 92.24	Mean = 95.96

**Table 2 ijms-25-01898-t002:** Primer sequences used in this study.

Gene	Forward Primer	Reverse Primer
*Shh*	TGAAAGCAGTGTGTCCAGCA	CGGTTGCCGGACTTCTACTT
*Smo*	AGAGTTCTACCTGTTTTAGCACCTG	TTACTACTCCGCCTCTTTCCAC
*Ptch1*	CCAACAACTACGCAAAAGCTA	TTTCTAATCGTCGGCACAAG
*Gli2/3*	GCCTGTGACAAACCTTGCAG	TCTGTCCCAAATGACCTGGC
*Bmp7*	TAATGTGAATGGGGCGAATG	TGGTGTAGTCCAAGCAGGGTC
*Bmpr1*	GAAGGCAGTTGGTTCAGGGA	GCTGTGTCCAGGATCCTGTC
*Bmp2/4*	CCGACCCGAAAGTTGAAGTG	TTTGCGGCTGTTGATTGC
*Bmpr2*	GGGACCATGATGCTGAAGCT	TGAGACAGACGGCTCCTGTA
*Smad2/3*	GTCACTTACAAGGAACCAGCATT	TGAACCCATCTACTGTCAACGAG
*Smad4*	GGAATGGAAGGGGCAAGTAG	ATGACAAGGGCTGTGGGGAC
*EF1a*	CACCACGAGTCTCTCCCTGA	GCTGTCACCACAGACCATTCC

## Data Availability

Data will be made available on reasonable request.

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
