# Peer review of "Transcriptomic Analysis Provides Insights into Candidate Genes and Molecular Pathways Involved in Growth of Mytilus coruscus Larvae"

_ijms, 2024, doi:10.3390/ijms25031898_

Round 1

Reviewer 1 Report (New Reviewer)

Comments and Suggestions for Authors

Positive comments to the authors:

1. The theme of the manuscript is particularly up -to -date as it is related to Transcriptomic Analysis of Mytilus Coruscus Larvae. This analysis of Provide Deep Insights Into the Molecular Basis of Physiological Adaptation, Metabolic Processes, and Growth Variability in Marine Bivalves;

2. The authors have carried out an extensive literary report related to the studied issue, with the study being scientific, as the Understanding the Mechanisms That Govern Growth and Development Across Various Stages Can Signostian Boostly Ereby Enhancing Aquaculture Efficiency and Advancing The Aquaculture Industry. Mytilus coruscus;

3. Very well formulated the purpose of the study, which meets the title of the manuscript and is associated with Identify Differentially Expressed Genes and Understand Their Expression Patterns, Enriching Understanding of the Genetic Foundations and Transc -Shelled Mussel Larvae;

4. The results are illustrated with 7 figures, some of which, however, are of an inappropriate scale and do not visualize the otherwise valuable data of the manuscript;

5. The material and methods section has been developed very well and describes the sampling, processing and mathematical analysis of the data obtained;

6. At the end of the manuscript, conclusions are formulated, which summarize the results obtained and the usefulness of the study in terms of biochemical ichthyogenetics. However, there are no clearly formulated recommendations for the aquacultural industry to emphasize the usefulness of the study as a scientific and applied development.

Negative notes and recommendations to the authors:

1. Some of the figures are cluttered with information and in the wrong scale. I recommend increasing the scale by 10-15% in Figures 2, 3, 4 and 7 for better visualization;

2. In the discussion section, the erudition of the authors in the field of biochemical ichthiogenetics and their ability to interpret the extensive data obtained. I recommend the authors to dig deeper into the bowels of molecular biology and to more fully use their experimental data to explain gene expression. They could even better emphasize the news and the two main signal paths established by them - Hedgehog Signal Road and TGF -Betta Signal Road - related to the growth and development of M. Coruscus larvae. In addition, the discovery of ten key growth-related genes should be exposed more extensively, each playing a crucial role in molecular function and regulating growth features in M. Coruscus.;

3. Every fundamental study such as the present has its applied aspects. I recommend the authors to find and dispose of better scientific and applied aspects of their study, which will be useful not only in the field of molecular biology, but also from a practical point of view of the aquacultural industry.

Comments on the Quality of English Language

Notes regarding the English language of the manuscript:

The manuscript is written in good and professional English. However, I recommend its final polishing by an English -speaking editor.

Author Response

Dear Editors and Reviewers:

Thank you for your comments on the paper. Efforts have been made to clarify the following points and correct the mistakes. The modifications based on the suggestions are explained in further detail below. The corresponding revised manuscript and annotated version with changes highlighted by blue color are submitted.

Negative notes and recommendations to the authors:

  1. Some of the figures are cluttered with information and in the wrong scale. I recommend increasing the scale by 10-15% in Figures 2, 3, 4 and 7 for better visualization;

Answer: The images placed in the article have been scaled and we have submitted the high-resolution images as attachments.

  1. In the discussion section, the erudition of the authors in the field of biochemical cytogenetics and their ability to interpret the extensive data obtained. I recommend the authors to dig deeper into the bowels of molecular biology and to more fully use their experimental data to explain gene expression. They could even better emphasize the news and the two main signal paths established by them - Hedgehog Signal Road and TGF -Betta Signal Road - related to the growth and development of M. Coruscus larvae. In addition, the discovery of ten key growth-related genes should be exposed more extensively, each playing a crucial role in molecular function and regulating growth features in M. Coruscus;

Answer: We have rewritten the discussion based on your advice (line 205-264).

  1. Every fundamental study such as the present has its applied aspects. I recommend the authors to find and dispose of better scientific and applied aspects of their study, which will be useful not only in the field of molecular biology, but also from a practical point of view of the aquacultural industry.

Answer: We have added related contents in discussion (line 265-280).

Reviewer 2 Report (New Reviewer)

Comments and Suggestions for Authors

This manuscript provides a comprehensive analysis of molecular and gene expression variations across five larval development stages of Mytilus           coruscus larvae , identifying differentially expressed genes, enriched pathways, and key growth-related genes. The authors also validated the findings using quantitative real-time PCR (qRT-PCR).

The work is intereting but could benefit by some improvements.

 The work is interesting , I enjoyed reading it but I have some comments and suggestions for improving it.

The introduction provides a solid foundation for the manuscript, effectively introducing the study's objectives, justifying its importance, and highlighting the key findings. With minor improvements, it could become an even stronger introduction that further engages the reader and sets the stage for the subsequent discussion and analysis.  More specifically, it could be strengthened by elaborating on the significance of signaling pathways and the potential applications of the findings.  For example, it would be better to expand on the role of Hedgehog and transforming growth factor-β signaling pathways in molluscan development, elaborate on the contribution of growth-related genes to larval development and briefly discuss the potential applications of the findings in aquaculture and larval health assessment.

Regarding the methodology,  I believe that you ould improve the clarity and reproducibility of their work by providing more detailed descriptions of the methods used. This includes providing comprehensive protocols for sample collection and RNA isolation (for example concentration/dilution, number of samples or pool of larvae and of how many  were sampled and used?).

 You could strengthen the discussion by clearly connecting their findings to existing knowledge in the field of molluscan development and molecular biology.
For example you
could deepen the discussion by exploring the biological implications of their findings. This could involve explaining how the identified gene expression patterns contribute to specific developmental processes, such as cell differentiation, tissue formation, or organogenesis. This would involve linking the identified genes and signaling pathways to specific cellular processes and regulatory networks.

You could acknowledge any limitations of their work and suggest potential areas for further research, by addressing potential controversies or alternative interpretations of their findings.

By incorporating these suggestions, you could transform the discussion from a mere summary of findings into a insightful and thought-provoking analysis that further advances our understanding of molluscan development and its underlying molecular mechanisms.

Author Response

Dear Editors and Reviewers:

Thank you for your comments on the paper. Efforts have been made to clarify the following points and correct the mistakes. The modifications based on the suggestions are explained in further detail below. The corresponding revised manuscript and annotated version with changes highlighted by blue color are submitted.

1.The introduction provides a solid foundation for the manuscript, effectively introducing the study's objectives, justifying its importance, and highlighting the key findings. With minor improvements, it could become an even stronger introduction that further engages the reader and sets the stage for the subsequent discussion and analysis. More specifically, it could be strengthened by elaborating on the significance of signaling pathways and the potential applications of the findings.  For example, it would be better to expand on the role of Hedgehog and transforming growth factor-β signaling pathways in molluscan development, elaborate on the contribution of growth-related genes to larval development and briefly discuss the potential applications of the findings in aquaculture and larval health assessment.

Answer: We have added some contents about the application of some growth and development pathways in Mollusca (line 71-87).

2.Regarding the methodology, I believe that you could improve the clarity and reproducibility of their work by providing more detailed descriptions of the methods used. This includes providing comprehensive protocols for sample collection and RNA isolation (for example concentration/dilution, number of samples or pool of larvae and of how many were sampled and used?).

Answer: About the collection of samples of mussels at each developmental stage, 500μL larvae samples stored in RNAlater solution were absorbed and about 30 larvae were placed under a 10x microscope (Different sizes of larvae at each stage were collected after density was adjusted in the breeding farm), so as to ensure adequate and equal larval samples for RNA extraction (line 287-290).

3.You could strengthen the discussion by clearly connecting their findings to existing knowledge in the field of molluscan development and molecular biology. For example, you could deepen the discussion by exploring the biological implications of their findings. This could involve explaining how the identified gene expression patterns contribute to specific developmental processes, such as cell differentiation, tissue formation, or organogenesis. This would involve linking the identified genes and signaling pathways to specific cellular processes and regulatory networks.

Answer: We have rewritten the discussion based on your advice (line 205-264).

4.You could acknowledge any limitations of their work and suggest potential areas for further research, by addressing potential controversies or alternative interpretations of their findings.

Answer: We have added related contents in discussion and conclusion (line 278-280 and 360-361).

This manuscript is a resubmission of an earlier submission. The following is a list of the peer review reports and author responses from that submission.

Round 1

Reviewer 1 Report

Comments and Suggestions for Authors

The authors report changes in gene expression through successive developmental stages of the bivalve, Mytilus coruscus, a major farmed species and important component of global food supply. The study is relevant to aquaculture practices in the context of efficacy and large scale production. Methods are explained sufficiently and data presentation is acceptable.

My broader problem with this paper is with the interpretation of the data and results. It is entirely expected that major differences in gene expression will be identified through developmental stages of any organism, and thus nothing reported here is particularly "remarkable", even though reference to "remarkable" is made at least 5 times in the manuscript, including in the Abstract. The focus on Hedgehog and TGF is certainly worth pointing out, but the textbook Figure (Fig. 5) presentation verbiage of these pathways is irrelevant and should be omitted. Likewise, most of the Discussion is pure speculation based on correlations.

It would be much more informative to replace the aforementioned Figure (Hedgehog, TGF signaling) with illustrations of the five developmental stages that were compared in the study. The nature of the analysis is limited to description, but the data can be overlayed onto the developmental stages, and ideally be connected to the broader importance of this cultured species in the context of food production.  This writing direction has been mentioned towards the end of the Discussion, but comprises only a minor component of the text. Other text, particularly in Discussion, should be minimized unless it can be connected directly to relevant aspects of the study (currently this is not the case). Comparisons to other developmental model organisms (e.g., evo-devo) would also be appropriate.

Comments on the Quality of English Language

English is ok but should be improved. Repetitive references to "remarkable" should be removed, as noted above, but other phrases are also repetitive (e.g., many sections open with "In this study...").

Capitalization of many words is inconsistent (e.g., Hedgehog vs hedgehog), and many words shouldn't be capitalized (e.g., in Abstract--spliceosome, apoptosis, etc.). Otherwise, check spelling and typos throughout (e.g., trochopho --> trochophore). 

Author Response

Dear Editors and Reviewers:

Thank you for your comments on the paper. Efforts have been made to clarify the following points and correct the mistakes. The modifications based on the suggestions are explained in further detail below. The corresponding revised manuscript and annotated version with changes highlighted by blue color are submitted.

1.My broader problem with this paper is with the interpretation of the data and results. It is entirely expected that major differences in gene expression will be identified through developmental stages of any organism, and thus nothing reported here is particularly "remarkable", even though reference to "remarkable" is made at least 5 times in the manuscript, including in the Abstract. The focus on Hedgehog and TGF is certainly worth pointing out, but the textbook Figure (Fig. 5) presentation verbiage of these pathways is irrelevant and should be omitted. Likewise, most of the Discussion is pure speculation based on correlations. It would be much more informative to replace the aforementioned Figure (Hedgehog, TGF signaling) with illustrations of the five developmental stages that were compared in the study. The nature of the analysis is limited to description, but the data can be overlayed onto the developmental stages, and ideally be connected to the broader importance of this cultured species in the context of food production. This writing direction has been mentioned towards the end of the Discussion, but comprises only a minor component of the text. Other text, particularly in Discussion, should be minimized unless it can be connected directly to relevant aspects of the study (currently this is not the case). Comparisons to other developmental model organisms (e.g., evo-devo) would also be appropriate.

Answer: We have written the discussion based on your advice.

2.Comments on the Quality of English Language

English is ok but should be improved. Repetitive references to "remarkable" should be removed, as noted above, but other phrases are also repetitive (e.g., many sections open with "In this study...").

Answer: We have corrected “remarkable” to “rapidly and powerful” in two locations. “In this study” was replaced to “during this experiment, based on this study and in this research”.

3.Capitalization of many words is inconsistent (e.g., Hedgehog vs hedgehog), and many words shouldn't be capitalized (e.g., in Abstract--spliceosome, apoptosis, etc.). Otherwise, check spelling and typos throughout (e.g., trochopho --> trochophore).

Answer: We have unified the capitalization of words and corrected the misspelling.

Reviewer 2 Report

Comments and Suggestions for Authors

I suggest that this paper can be accepted after minor revision. For further information, I have attached the manuscript containing all my observations made directly in the text.

Comments on the Quality of English Language

The present language quality is well and solely minor revision needs  to be improved.

Author Response

Dear Editors and Reviewers:

Thank you for your comments on the paper. Efforts have been made to clarify the following points and correct the mistakes. The modifications based on the suggestions are explained in further detail below. The corresponding revised manuscript and annotated version with changes highlighted by blue color are submitted.

  1. biological > biology

Answer: We have corrected biological to biology.

  1. Please avoid using the duplicate words in academic writing. So, you can write investigated instead of explored

Answer: We have corrected explored to investigated.

  1. delete "the"

Answer: We have deleted.

  1. Which program has been utilized to conduct PCA? The programmes that were used to conduct statistical analyses in this study, must be cited.

Answer: “Performing Principal Component Analysis (PCA) using origin software and visualized.” This content is presented in the materials and methods section.

  1. The font size has to be changed in order to improve the legibility of the legend images.

Answer: The images shown in the article are compressed, clearer images have been uploaded separately.

6.In general, this manuscript has to include a section on statistical analysis to clarify how the authors handled data analysis, otherwise it will be incomplete structure?

Answer: Statistical analysis content is presented in the section 4 materials and methods.

7.The same comment on the Figure 2.

Answer: Statistical analysis content is presented in the section 4 materials and methods.

Round 2

Reviewer 1 Report

Comments and Suggestions for Authors

The authors have made a minimal attempt to address raised concerns. Some textual updates are flawed, e.g., "...with a rapidly average annual growth rate..." in the opening paragraph of Introduction. Discussion is slightly more relevant than previous version, but paragraphs are awkwardly long and information lacks cohesion. No attempt is made to address major concerns regarding Fig. 5 (should be deleted) and presentation schematic of developmental stages (should be added). 

Comments on the Quality of English Language

Needs editing

Author Response

Dear Editors and Reviewers:

Thank you for your comments on the paper. Efforts have been made to clarify the following points and correct the mistakes. The modifications based on the suggestions are explained in further detail below. The corresponding revised manuscript and annotated version with changes highlighted by blue color are submitted.

  1. The authors have made a minimal attempt to address raised concerns. Some textual updates are flawed, e.g., "...with a rapidly average annual growth rate..." in the opening paragraph of Introduction. Discussion is slightly more relevant than previous version, but paragraphs are awkwardly long and information lacks cohesion. No attempt is made to address major concerns regarding Fig. 5 (should be deleted) and presentation schematic of developmental stages (should be added).

Answer: We have deleted Fig. 5 and rewritten the discussion based on the developmental stages.